# Role of Immunotherapy in Sarcomas

**DOI:** 10.3390/ijms25021266

**Published:** 2024-01-19

**Authors:** Shivani Dalal, Khine Swe Shan, Nyein Nyein Thaw Dar, Atif Hussein, Alejandra Ergle

**Affiliations:** Memorial Healthcare, Division of Hematology and Oncology, Pembroke Pines, FL 33028, USA; kshan@mhs.net (K.S.S.); nthawdar@mhs.net (N.N.T.D.); ahussein@mhs.net (A.H.); aergle@mhs.net (A.E.)

**Keywords:** immunotherapy, sarcomas, adoptive cell therapy, CAR T-cell therapy, oncolytic viruses, TCR therapy, immune microenvironment

## Abstract

Sarcomas are a group of malignancies of mesenchymal origin with a plethora of subtypes. Given the sheer heterogeneity of various subtypes and the rarity of the disease, the management of sarcomas has been challenging, with poor patient outcomes. Surgery, radiation therapy and chemotherapy have remained the backbone of treatment in patients with sarcoma. The introduction of immunotherapy has revolutionized the treatment of various solid and hematological malignancies. In this review, we discuss the basics of immunotherapy and the immune microenvironment in sarcomas; various modalities of immunotherapy, like immune checkpoint blockade, oncolytic viruses, cancer-targeted antibodies, vaccine therapy; and adoptive cell therapies like CAR T-cell therapy, T-cell therapy, and TCR therapy.

## 1. Introduction

Sarcomas are a group of heterogeneous malignancies of mesenchymal origin with more than 100 histologic subtypes. They have diverse molecular, genetic, and clinical features and comprise 1% of adult malignancies [1,2,3]. They can be generally classified into two major types: soft-tissue sarcomas (STSs), which include more than 50 subtypes (with the most common being liposarcoma, leiomyosarcoma and undifferentiated pleomorphic sarcoma), and bone sarcomas (BSs) (osteosarcoma, chondrosarcoma, and Ewing sarcoma). Some sarcoma subtypes are extremely rare and hence are underrepresented in clinical trials.

Recently, the incorporation of immunotherapies into treatment regimens has been heavily investigated and has revolutionized the treatment of solid tumors. Immune checkpoint inhibitors (ICIs) have limited efficacy in sarcomas compared to other solid tumors; however, they have shown some activity in certain subtypes. Moreover, clinical trials are heterogeneous, as most have been basket trials with a variety of different sarcoma subtypes despite their unique biological characteristics, thus making it difficult to utilize ICIs in rare subtypes. There are several challenges with immunotherapy use in sarcoma due to tumor heterogeneity, the paucity of targetable antigens in sarcoma subtypes by therapeutic antibodies, vaccines, and chimeric antigen receptors, and the lack of individualized trials for rare subtypes. However, the combination of ICIs with other therapies appears to have synergistic effects, and potential treatment options such as adoptive cell therapy and oncolytic viruses are emerging. In this review, we discuss the biological basis, current clinical trials, and future challenges of immunotherapy in advanced sarcomas.

## 2. Current Sarcoma Treatment Landscape

The treatment of STSs can be based on the given subtype, such as STSs of extremity, superficial/trunk or head and neck; retroperitoneal or intra-abdominal STSs, desmoid tumors, and rhabdomyosarcomas [4]. Treatment requires evaluation and management by a multidisciplinary team, including experienced pathologists, radiologists, medical oncologists, surgical oncologists, and radiation oncologists, for the consideration of systemic therapy, surgery, and/or radiation [4]. Conventional chemotherapy, including anthracycline-based regimens, is the standard of care for most advanced and metastatic STSs, and non-anthracycline-based regimens are preferred in angiosarcomas (ASs) and perivascular epithelioid cell neoplasms [5,6,7]. Several tyrosine kinase inhibitors (TKIs) have shown promising results in specific histologic subtypes of advanced and metastatic STS. Pazopanib, a multitargeted tyrosine kinase inhibitor (multi-TKI), can be used as a single agent in metastatic non-lipogenic STS patients previously treated with anthracycline-based regimens, or as a front-line treatment in advanced and/or metastatic STS patients who are not candidates for anthracycline-based regimens [8,9]. Other TKIs used in advanced STS include regorafinib (non-adipocytic sarcoma and AS), sorafenib (desmoid tumors) and imatinib (dermatofibrosarcoma protuberans and gastrointestinal stromal tumors (GISTs)) [10,11,12,13].

The treatment of BSs can be vastly different depending on the subtype. Osteosarcomas (OSs) are usually radiation-resistant, and treatment involves wide excision with perioperative chemotherapy including doxorubicin, cisplatin, and high-dose methotrexate [14]. On the other hand, Ewing sarcomas (ESs) are sensitive to radiation, and treatment usually involves perioperative chemotherapy with vincristine, doxorubicin, cyclophosphamide, ifosfamide and etoposide with surgery with or without radiation [15]. Chondrosarcomas (CSs) are chemotherapy- and radiation-resistant, and the primary treatment is surgical resection [16]. Recently, TKIs like regorafenib, cabozantinib and apatinib have also been shown to be effective in OS [17]. The investigation of TKIs for other BSs has not been well developed, but it has shown encouraging results in preclinical and early trials in ESs and chondrosarcomas [17]. Currently, atezolizumab is the only immunotherapy drug approved by the Food and Drug Administration (FDA) for sarcomas, and it was approved for unresectable or metastatic alveolar soft part sarcoma (ASPS) on 9 December 2022 [18]. Pembrolizumab can be considered as a second-line treatment for patients with certain subtypes of advanced or metastatic STS, including myxofibrosarcoma (MFS), undifferentiated pleomorphic sarcoma (UPS), cutaneous AS and undifferentiated sarcomas [4].

## 3. Cancer Immunotherapy

The clinical benefits of immune enhancement in cancers have been well-proven since the 1800s. Immunotherapy is the fifth pillar of cancer treatment after surgery, chemotherapy, radiation therapy and targeted therapy. In some cases, it has become the first line of treatment [19]. Cancer immunotherapies can be classified based on the mechanism of action: (i) a checkpoint blockade that removes the natural inhibitory signals of the immune system: a CTLA-4 inhibitor (Cytotoxic T-lymphocyte-associated protein) (e.g., ipilimumab), a PD-1 inhibitor (Program Cell Death protein 1) (e.g., pembrolizumab, nivolumab), or a PD-L1 inhibitor (Programmed Death ligand 1) (e.g., atezolizumab, avelumab, durvalumab); (ii) adoptive cell therapies including the infusion of modified immune effector cells (T-cells, Chimeric Antigen Receptor T-cells (CAR T-cells), NK cells, or TCR based therapy; (iii) cancer vaccines; (iv) oncolytic viruses; and (v) cancer-targeted antibodies. The success of immunotherapy is significantly affected by the immunogenicity of the tumor, the tumor mutation burden and the tumor microenvironment.

## 4. Immune Microenvironment and Biomarkers in Sarcoma

The tumor microenvironment (TME) is a primary location for cell-to-cell interactions around the tumor, signal transfer or delivery, and cytokine production. It plays an important role in tumor cells escaping the natural immune system and can similarly affect the efficacy of some immunotherapies. The components of the TME are tumor-associated macrophages (TAMs); pro-inflammatory cytokines; other immune checkpoint modulators; regulatory T-cells (Treg); immunosuppressive cytokines such as transforming growth factor-beta (TGF-beta); pro-angiogenic cytokines like fibroblast growth factor (FBGF); or vascular endothelial growth factor (VEGF). They function through a complex pathway to maintain tumor growth and overcome the anti-cancer immune system.

Being a heterogeneous disease with multiple subtypes, sarcoma has a variety of genetic profiles and characteristics of tumor cells in each individual subtype [20]. The tumor mutational burden (TMB) and microsatellite instability (MSI) status have commonly been used to predict tumor response to immunotherapy. There is high variability of TMB among different subtypes of sarcoma. For example, a few sarcoma subtypes such as soft-tissue rhabdomyosarcoma, alveolar, liposarcoma, and synovial sarcoma have a low TMB, whereas soft-tissue angiosarcoma has a high TMB with a median mutational burden of 3.8 mutations/Mb, and 13.4% of cases have more than 20 mutations/Mb [20].

Another potential predictive factor used to evaluate the utility of immunotherapy in sarcoma treatment is tertiary lymphoid structures (TLSs), which refer to the organized aggregates of lymphoid cells forming around the tumor cells. Usually observed via the use of immunohistochemistry (IHC), the aggregation of B cell follicles, dendritic cells, helper T-cells (CD4+) and cytotoxic T-cells (CD8+) represents the TLS phenotype, which is frequently found in immune-high sarcoma types and can predict better outcomes with immune checkpoint inhibitors (ICI) [21,22]. This hypothesis is supported in the PEMBROSARC trial, a multicohort phase II trial, which showed that the presence of TLS features in the TME was associated with a higher pembrolizumab treatment response [23]. Interestingly, a high infiltration of regulatory T (Treg) cells, which modulate the immune function, was found in TLS-positive non-responder groups and decreased the effect of pembrolizumab [23]. As such, the clinical impact of TLSs is still controversial, and the known predictive value is currently limited.

By analyzing bulk RNA transcriptome data from tumor-infiltrating lymphocytes (TILs) in 85 osteosarcoma patients, researchers identified 5 different TIL marker genes. These genes were used to create a risk model with both prognostic and predictive value. In this model, varying levels of expression of these five genes were used to classify patients based on higher survival (“low risk”) or lower survival (“high risk”). Additionally, it was found that high-risk tumors had a lower abundance of immune cell infiltration, whereas low-risk tumors had a higher expression of immune checkpoint genes such as CTLA4 and LAG3, which could provide a positive predictive value in the response to immunotherapy for low-risk patients [24]. The combination of IHC for the TLS phenotype and the TIL molecular RNA signature could potentially be used to provide enhanced prognostic and predictive models.

PD-L1 expression varies in different subtypes of tumors. High PD-L1 expression is found in high-grade dedifferentiated leiomyosarcoma [25]. However, there is no sufficient data to support PD-1 or PD-L1 as a predictive biomarker for ICI treatment. The efficacy of pembrolizumab was not related to the level of PD-L1 expression in the SARC028 trial [26]. In endometrial sarcoma, PD-L2 expression was associated with mismatch repair (MMR)-proficient tumors and lower OS rates when compared to PD-L1 expression [27]. Similar findings were reported in uterine adenosarcoma, which also showed that PD-L1 expression did not correlate with the density of TILs, but PD-L2 expression is positively correlated with TP53 mutation, which is associated with worse clinical outcomes [28].

## 5. Immune Checkpoint Blockade

ICIs target tumor cells’ known inhibitory signals to T-cells and have shown responses in solid tumors. Immune checkpoint receptors, including CTLA-4, PD-1, PD-L1, and LAG-3 (Lymphocyte activation gene 3), are inhibitory molecules present on the surface of immune cells, cancer cells and other supporting cells in the TME [29]. Sarcomas usually have a low TMB, an immunosuppressive TME, and low PD-L1 expression, and only a few percent of these tumors are mismatch repair-deficient; they are not considered immune-sensitive tumors [1]. Even though the response to ICIs in sarcomas is not high in general, there is some benefit in specific histological subtypes [1]. In addition, the role of PD-L1 expression in STS is unclear, as responses are seen even in cases without PD-L1 expression [30].

Earlier trials with single-agent immunotherapy (anti-PD-1 nivolumab, anti-CTLA-4 ipilimumab) failed to demonstrate significant antitumor activity [31,32]. One of the first ICI trials with positive results was the prospective single-arm phase II trial SARC028, evaluating the anti-PD1 pembrolizumab as a second-line treatment in 80 patients with either STS or BS [30]. This study demonstrated an objective response rate (ORR) of 18% in patients with STS with a median progression-free survival (PFS) and overall survival (OS) of 18 and 49 weeks, respectively [30]. One patient with UPS had a complete response (CR). The benefit was limited to the patients with UPS and de-differentiated liposarcoma (DDLPS), and a minimal benefit was seen in synovial sarcoma (SS), leiomyosarcoma (LMS) and BS [30]. However, this trial excluded rare STS subtypes; thus, the efficacy of anti-PD1 in rare STS subtypes was not evaluated. The response of UPS to pembrolizumab was further confirmed in an expansion cohort of SARC028 with two CRs and seven partial responses (PRs) in the UPS cohort; however, a response was not seen in the liposarcoma (LPS) cohort [33].

A phase II French AcSé trial, evaluating the efficacy of pembrolizumab in different cohorts of patients with rare cancers, including the rarest sarcoma subtypes, showed an ORR of 15.3% with a disease control rate (DCR) of 52.5% [34]. It demonstrated the highest response rates in ASPS, with a 50% ORR, and in SMARCA4-deficient malignant rhabdoid tumors (SMRTs), with a 27% ORR. Other response rates were 8.8% in chordoma, 12.5% in desmoplastic small round-cell tumors (DSCRTs) and 3.2% in other histotypes [34].

A pooled analysis of several clinical trials investigating anti-PD-1/PD-L1 immunotherapy in advanced STS, including UPS, LPS, LMS, and ASPS, reported that among 384 patients, 39.8% received anti-PD1/PD-L1 immunotherapy and had an ORR of 15.1% and median PFS of 58.5% [26]. ASPS and UPS were among the highest responders (48.4% and 15.7%, respectively) and LPS and LMS were among the lowest (7.3% and 6.9%, respectively) [35]. A retrospective study of nivolumab with or without ipilimumab was evaluated in PD-L1-positive STS in the first-line setting, and the study demonstrated an ORR of 13% in the combination group vs. 7% in the nivolumab group [36].

Targeting various immune checkpoints, including PD-1, CTLA-4 and LAG-3, simultaneously is a promising approach to improving the efficacy of immunotherapy. Lussier et al. showed that CTLA-4 expression was upregulated in T-cells infiltrating PD-L1 antibody-resistant tumors in mice with metastatic OS, suggesting a potential synergic effect of an anti-CTLA-4 and PD-L1 blockade [37]. Combination checkpoint inhibition with nivolumab and ipilimumab was evaluated in previously treated patients with advanced STS in the phase II Alliance A091401 trial, exhibiting an ORR of 16% and a median PFS and OS of 4.1 and 14.3 months, respectively. Better responses were seen with combination therapy, with the best responses being in UPS (33%), LMS (14.2), and AS (33%) [38].

A dual blockade of PD-1 and LAG-3 has also shown synergistic antitumor activity in preclinical models. A phase II basket trial of anti-PD1 spartalizumab plus anti-LAG3 LAG525 with a cohort of 10 sarcoma patients reported 40% DCR at 24 weeks [39]. The expansion criterion was not met, but the sarcoma cohort was not found to be futile [39]. Currently, nivolumab plus anti-LAG3 relatimab vs. nivolumab alone is being investigated in advanced STS in the ongoing phase II clinical trial NCT04095208 (CONGRATS trial).

A tumor biomarker analysis of ASPS affirms the presence of PD-1/PD-L1 immune checkpoint components, suggesting that immune checkpoint inhibition could be beneficial in advanced ASPS [40]. A phase II trial of 43 evaluable patients with ASPS using the anti-PD-L1 atezolizumab showed an ORR of 37.2% with a median duration of response (DOR) of 16.5 months [40]. Recently updated results with 52 patients confirmed the ORR of 37% with a DOR of 24.7 months and a PFS of 20.8 months [41]. Atezolizumab obtained FDA approval for unresectable or metastatic ASPS on 9 December 2022 [18].

Given the lack of randomized phase III trials and limited therapeutic alternatives for patients who progressed while undergoing chemotherapy treatment, ICIs could be considered, especially in patients with UPS, DDLPS and ASPS. Dual immune blockades appear to show higher response rates and may be considered in selected patients. Further details and a summary of ICI trials are listed in Table 1.

## 6. Combination of ICIs with Tyrosine Kinase Inhibitors

The quest to find the optimal response of ICIs in sarcomas has led to the combination of ICIs with targeted therapies, mainly anti-angiogenic and multi-TKIs. Preclinically, the normalization of abnormal tumor vessels and the increased infiltration of immune effector cells into tumors by anti-angiogenic TKIs have been shown to enhance the efficacy of ICIs [68]. In addition to blocking the immune-suppressive effect of vascular epidermal growth factors (VEGFs), multi-TKIs seem to have a favorable immune modulating effect by decreasing the arrival of myeloid-derived suppressor cells and tumor-associated macrophages and increasing the infiltration of dendritic cells, natural killer cells and CD8+ lymphocytes, potentially making the combination of multi-TKIs with ICIs reasonable [69].

Nivolumab and sunitinib in combination were tested in advanced BS in the phase I/II IMMUNOSARC trial, with an ORR of 5%, one CR in dedifferentiated chondrosarcoma (DDCS), one PR, twenty-two SDs, and a PFS of 3.7 months [50]. In this same trial, an advanced STS cohort was also evaluated and found to have an ORR of 21%, 1 CR in AS, 5 PRs, and 33 SDs among 46 patients, and a median PFS of 5.6 months [51]. Pembrolizumab with axitinib demonstrated promising responses in a phase II trial of 33 advanced sarcoma patients among which 51% were previously treated [48]. An ORR of 25% (8 PRs) and a PFS of 4.7 months was noted, and the most benefit seen in ASPS, with a 50% ORR and a PFS of 12.4 months [48]. The benefit was thought to be due to high TIL and PD-L1 expression in ASPS tumors [48]. Another anti-PD1, camrelizumab, was evaluated with apatinib in the phase II APFAO trial of 43 patients with chemotherapy-refractory OS. The study showed an ORR of 20.9% and a PFS of 6.2 months, with the longest PFS observed in patients with lung metastasis and a PD-L1 tumor proportion score of 5% [49].

The combination of durvalumab with pazopanib in previously treated patients with advanced STS showed an ORR of 28.3% and a median PFS of 7.7 months, with objective responses in ASPS, AS, UPS, and dedifferentiated chondrosarcoma (DSRCT) [52,70]. Upon further analysis, tumors with high CD20+B cell infiltration and vessel density were reported to have a longer PFS and a better response than those without [70].

Combining ICIs and TKIs is a potential option in patients with advanced STS after progression on standard chemotherapy. Unfortunately, the effectiveness of this combination in BS is not very promising. Further investigation is necessary to compare the benefits of these combinations with single ICI or TKI monotherapy, including the utility in BS. Also, they can be potentially considered to be used as a front-line treatment in selected patients unfit for anthracycline-based chemotherapy. 

## 7. Combination of ICIs with Conventional Chemotherapy

The rationale for the combination of chemotherapy with ICIs is that the induction of cell death by chemotherapy can potentiate immunotherapy response by exposing cellular debris to the immune cells [71]. Cytotoxic drugs can result in DNA damage, leading to cell death, the release of immunostimulatory signals known as damage-associated molecular patterns (DAMPs), and proteins that work as “danger signals”, with the eventual upregulation of PD1 and the enhancement of effector lymphocytes activity [17].

The phase II PEMBROSARC trial of pembrolizumab with cyclophosphamide in 50 patients with locally advanced or metastatic sarcomas (both treatment-naive and pretreated patients), including LMS, UPS and GIST, showed limited activity, with just one PR in a patient with solitary fibrous tumor (SFT) with PD-L1 expression greater than 10% in immune cells [56]. This trial showed that the clinical benefit of ICIs with chemotherapy is very limited in an unselected population. An amended study of PEMBROSARC, which included 35 patients with TLS-positive advanced STS, showed a 6-month non-progression rate and an ORR of 40% and 30%, respectively, compared with 4.9% and 2.4%, respectively, in the previous cohort of the PEMBROSARC study [23]. Thus, the presence of TLSs in advanced STS could be a potential predictive biomarker used to improve the selection of patients for the treatment of ICIs with chemotherapy.

The first combination phase I/II trial of pembrolizumab with doxorubicin in 37 anthracycline-naive patients with advanced STS demonstrated an ORR of 19% (7 PRs), a PFS of 8.1 months and an OS of 27.6 months, with more prominent results in UPS and DDLPS subtypes [64]. Even though the study failed to meet its primary endpoint, with a response rate of 29%, combination therapy was associated with longer PFS than doxorubicin alone (8.1 months vs. 4.1 months) [64]. Another trial of pembrolizumab and doxorubicin in advanced STS showed that patients with PD-L1 ≥ 5% had a three times greater ORR (63.6%) than those with PD-L1 < 5% [65]. In this study, PD-L1 expression was not associated with improved PFS or OS but was associated with improved ORR [65].

LMS and LPS are usually resistant to PD-1/PD-L1 inhibition, likely due to the infiltration of high levels of immunosuppressive TAMs [72]. Trabectedin could influence TME and reduce TAMs, thus improving antitumor adaptive immunity to anti-PD1 therapy [62]. A combination of Trabectedin and anti-PD-L1 avelumab in a phase I/II study of patients with advanced LPS and LMS showed 2 PRs and 11 SDs, with a 6-month PFS of 50.1% [72]. In the phase II NiTraSarc trial, a nivolumab with trabectedin combination was evaluated as a second-line treatment in anthracycline pretreated advanced STS patients (Group A with advanced LPS or LMS and Group B with other sarcomas, including pleomorphic, spindle cell, fibromyxoid, synovial and epithelial sarcoma) [62]. In the late combination cohort (LCC), patients are treated with three cycles of trabectedin followed by trabectedin plus nivolumab, whereas the early combination cohort (ECC) started combination treatment at cycle 2 [62]. After a median follow-up of 16.6 months, the PFS in Group A was 47.6% (60% in LCC vs. 36.4% in ECC), and it was 14.6% in Group B. The median PFS was higher in Group A compared to Group B (5.5 vs. 2.3 months) and longer in LCC vs. ECC (9.8 vs. 4.4 months) [62]. OS was much higher in Group A vs. Group B (18.7 vs. 5.6 months), and longer in LCC vs. ECC (24.6 vs. 13.9 months) [62]. This confirmed the activity of trabectedin followed by a combination with nivolumab in LPS and LMS [62].

Results of a phase II trial of doxorubicin with anti-CTLA-4 zalifrelimab and anti-PD1 balstilimab as the first- and second-line treatment in 28 patients without prior doxorubicin or ICIs were recently reported [66]. The study had a two-stage design, with stage 1 comprised of a priming dose of zalifrelimab and balstilimab in cycle 1 prior to adding doxorubicin at cycle 2, and stage 2 comprised of giving all drugs at cycle 1. The ORR was 36% and the DCR was 86%, with a DOR of 12.8 weeks in overall population [66]. Patients who received ICI priming at cycle 1 prior to chemotherapy had a better 6-month PFS (56.3 vs. 25%), ORR (56% vs. 8.3%) and DCR (94% vs. 75%) compared to those in stage 2 [66].

The ICI and chemotherapy combination seems to respond better in specific histologic subtypes such as LPS, LMS and UPS. However, it is difficult to obtain meaningful results given the heterogeneity in the selections of patients and difficulty confirming the therapeutic benefit of immunotherapy compared to chemotherapy without randomized trials. Moreover, further investigations are needed to evaluate the sequence of priming with either ICIs or chemotherapy to find the most effective treatments. Phase III trials comparing these regimens to the standard of care are necessary to confirm these findings. Moreover, this combination is not very well investigated in BS. 

## 8. Combination of Immunotherapy with Local Radiation Therapy

Radiation may produce neoantigens that enhance the immunogenicity of tumors with a low TMB, making them more responsive to ICIs in a T-cell-dependent manner [1]. A randomized phase II non-comparative trial evaluating neoadjuvant radiation with nivolumab alone or nivolumab with ipilimumab in 24 surgically resectable patients with DDLPS or UPS showed significant clinical activity in UPS, with a median pathological response of 95%. This figure was 22.5% in the DDLPS cohort, with responses being similar irrespective of the addition of ipilimumab [73]. It was found that radiation therapy in UPS increased tumor-infiltrating immune cells and tumor PD-L1 expression [73,74]. The combination of ICIs with radiation therapy is currently being investigated in several trials (see Table 2).

It is difficult to determine the most immune-sensitive sarcoma subtypes given the heterogeneity of sarcomas, the limited numbers of patients enrolled, the inconsistencies in the designs and results of various trials, the lack of phase III randomized clinical trials, the lack of representatives of rare histology subtypes, and the lack of validated biomarkers for ICIs. Currently, there is a need for trials with better designs and individualized studies investigating each group of sarcomas that share common biological characteristics.

## 9. Adoptive Cell Therapy

Adoptive cell therapy (ACT) is a new and innovative strategy that uses immunological principles to target cancer cells. It has the potential to induce a durable response in tumors, and promising results have been seen in hematological malignancies and some solid tumors. ACT involves the extraction of immune cells from a patient’s blood, tumor tissue or healthy donor via leukapheresis. The cells are then genetically engineered ex vivo to make them targeted toward specific tumor cells and then expanded prior to reinfusion into the patient. T-cells are capable of killing tumor cells directly and activating additional immune cells, subsequently eliciting an immune response. Three classic examples of ACTs used for cancer immunotherapy are:A.T-cell therapy.B.Chimeric antigen receptor T-cell therapy (CAR-T).C.T-cell receptor-based therapy (TCR).
(1)T-Cell Therapy

T-cell-based therapy is comprised of TILs, which are extracted from the tumor, activated ex vivo, expanded and reinfused in the patient along with immune enhancing adjuvants, such as interleukin-2 (IL-2), to induce a durable immunological response against the tumor cells. The patient receives a lymphodepleting chemotherapy regimen, such as cyclophosphamide or fludarabine, prior to the reinfusion of these TILs, to deplete the innate T-cells that may suppress the proliferation of the infused T-cells in the body [75]. In contrast to engineered TCRs and CAR-T-cells, this is the only ACT technique with multiple T-cell receptor clones able to target the antigenic heterogeneity of sarcoma [76]. TILs in cancer immunotherapy have been studied in various cancers, such as renal cell carcinoma, breast cancer, colon cancer, and melanoma, amongst various others [77]. The earliest studies, dating back three decades, by Balch et al. reported that TILs were present in about 35% of patients with sarcoma, particularly gastrointestinal stromal tumors (GISTs), STS, Ewing sarcoma (ES), osteosarcoma and uterine sarcomas; however, their potential consideration as predictive markers is unclear based on the current data [77]. This approach of infusing TILs expanded ex vivo was found to have remarkable efficacy in melanoma, with durable response rates and long-term survival benefits [78].

Mullinax at al. conducted a study on 70 patients with STS and demonstrated the feasibility of creating a TIL culture. The study showed that TILs demonstrated tumor-specific reactivity through a IFNγ release assay in 51 samples. The tumor-specific activity was noted in 56.3% of patients (9/16) using the fragment method (tumor fragments minced into pieces ~1 mm^3^ in size) and in 40% (14/35) using the digest method (tumor tissue processed into a single cell suspension using both mechanical and enzymatic disruption) (*p* = 0.37 comparing fragment vs. digest methods) [79]. In a retrospective study conducted by Zhou et al., 60 patients with chemotherapy-resistant metastatic OS were enrolled, and a combined approach with adoptive TIL and anti-PD1 therapy was investigated. The results were encouraging, with an ORR of 36.7%, a DCR of 80%, and a median PFS of 5.8 months. OS was 23.7 months in responders versus 8.7 months in non-responders (*p* < 0.0001) [80].

However, despite promising preclinical and retrospective data, further research is encouraged to understand and navigate the challenges still faced by TIL therapy, especially in sarcomas, given the substantial heterogeneity between different subtypes.
(2)CAR T-Cell Therapy

CAR (chimeric antigen receptor) T-cell therapy is a type of adoptive cell therapy that aims to modify the DNA of a patient’s T-lymphocytes in order to enable them to selectively target and eliminate cancer cells. The identification of tumor-specific antigens for CAR T-cell targeting is challenging in solid tumors given their intense antigenic heterogeneity. Due to their polyclonal expansion and accumulative mutations, it is hard to find homogeneously expressed targets. See Figure 1 for CAR T-Cell production process.

A chimeric antigen receptor structure consists of: (See Figure 2 for CAR T-cell structure)
An antigen-recognition domain—a single-chain variable fragment (scFv) as a part of a genetically engineered monoclonal antibody that targets the tumor antigen.A hinge that links a recognition site to the transmembrane domain bridging the membrane.An intracellular domain that facilitates T-cell receptor signaling [81].

The positive results noted in clinical trials using CAR T-cell therapy in B-cell lymphomas and acute lymphoblastic leukemias led to the extension of the study of CAR T-cells in the treatment of various types of sarcomas. GD2 (diasialoganglioside) is an attractive target for cancer immunotherapy as it is over-expressed on various tumors, including neuroblastoma, melanoma, OS, ES, and RMS, while it is rarely expressed in normal tissue. T-cells expressing the first-generation anti-GD2 chimeric antigen receptors (CARs) were safe and had modest antitumor activity in some patients with refractory neuroblastoma [82]. Clinical trials testing the use of anti-GD2 CAR T-cells in patients with sarcomas and other GD-2-positive solid tumors are currently ongoing (see Table 3).

Another important phase I/II trial tested escalating doses of T-cells expressing a HER2-specific CAR in patients with recurrent/refractory HER2-positive sarcoma [83]. This study demonstrated that the CAR T-cells could persist for 6 weeks without major toxicities. This has set the stage for ongoing studies that combine anti-HER2 CAR T-cells with other immunomodulatory approaches to enhance CAR T-cell expansion and persistence [84]. Another important phase I clinical trial in sarcoma is aimed at testing the combination of anti-HER2 CAR T-cell therapy in combination with immune checkpoint blocking agents such as pembrolizumab or nivolumab (NCT04995003). These patients are typically pretreated with lymphodepleting agents such as cyclophosphamide and fludarabine prior to an infusion of CAR T-cells targeting the HER2 receptor. One week after the patient receives the HER2 CAR T-cells, they will begin treatment with pembrolizumab every three weeks or nivolumab every two weeks. This study is currently active and recruiting.
(3)TCR Therapy

T-cell receptor-based therapy utilizes engineered T lymphocytes specifically targeted towards surface tumor antigens. T-cell receptor (TCR)-engineered effector cells use a naturally occurring TCR, in contrast to CAR T-cell technology, which uses a foreign receptor introduced into the immune effector cells that helps recognize tumor cell surface proteins [85]. See Figure 3 for TCR structure.

In this strategy, the patient’s autologous T-cells are extracted through leukapheresis or from tumor tissue. These cells are then modified ex vivo through a lentivirus or retrovirus vector encoding a specific TCR gene and expanded prior to the reinfusion of cells into the patient. TCR therapy recognizes fragments of tumor-specific antigens that are presented by MHC molecules on tumor cell surface. The binding of TCRs to the MHC–antigen complex leads to the activation of T lymphocytes. T-cells can kill tumors directly and attract additional immune cells, thereby eliciting an immune response. It is crucial to identify tumor-specific antigens that are overexpressed in solid tumors with absent or limited expression in normal tissues. The expression of cancer testis antigens (CTAs), including melanoma antigen gene (MAGE), synovial sarcoma X (SSX) and New York esophageal squamous cell carcinoma gene-1 (NY-ESO-1), is restricted to the germline in normal tissue, but these molecules are broadly upregulated in various tumors. The expression of either NY-ESO-1 and/or MAGE-A4 has been observed in more than 50% of primary synovial sarcoma specimens. They are also observed in myxoid liposarcoma, osteosarcomas, pleomorphic liposarcoma and chondrosarcomas, making them appealing targets for TCR-based therapies [86,87,88].

In an interesting phase I/II study by Ramachandran et al., patients with advanced synovial sarcoma were injected with genetically modified autologous T-cells expressing NY-ESO1-1c259, an anti-NY-ESO-specific receptor. qPCR was used to determine engineered T-cell persistence, and immunoassay was used to evaluate serum cytokines. Transcriptomic analyses and immunohistochemistry were performed on tumor biopsies from patients before and after T-cell infusion. Of the 42 patients that were evaluated, 1 patient achieved a complete response, 14 achieved partial responses, and 24 showed stable disease (SD), and progressive disease (PD) was observed in only 3 patients. The study concluded that a lymphodepletion regimen containing high doses of fludarabine and cyclophosphamide is necessary for genetically modified autologous T-cell persistence and efficacy [89]. Another important pilot trial by Robbins et al. tested autologous TCR-transduced T-cells following lymphodepleting chemotherapy on patients with metastatic synovial sarcoma or melanoma expressing NY-ESO-1 that were refractory to standard treatment regimens. Out of 18 patients with NY-ESO-1-positive synovial sarcomas, 11 had objective clinical responses. The estimated overall 3- and 5-year survival rates for patients with synovial sarcoma were 38% and 14%, respectively [90]. A phase I clinical trial by Pan et al. enrolled 12 patients (10 patients with synovial sarcoma and 2 with liposarcoma) with advanced, unresectable sarcoma and HLA-A*02:01 and NY-ESO-1 expression. These patients received a lymphodepleting chemotherapy regimen with cyclophosphamide and fludarabine. Autologous T-cells engineered to express a high-affinity NY-ESO-1-specific TCR derived from peripheral blood mononuclear cells of a healthy HLA-matched donor were injected into the patients followed by a low dose of interleukin 2 usage to minimize toxicity. There were no serious adverse events in the 12 patients that were enrolled. Currently, a phase II study is ongoing to assess the safety and efficacy of this drug in patients with advanced soft-tissue sarcoma [91]. Another promising phase I Japanese study in the field of TCR therapy for sarcomas is one by Kawai et al., which studied the safety and efficacy of infusion of autologous T lymphocytes expressing NY-ESO antigen-specific TCR in patients with advanced or recurrent synovial sarcoma who are resistant to anthracycline and not surgical candidates [92]. When compared to pazopanib, the efficacy of treatment in terms of overall response rate and overall survival was strongly superior. The safety profile was acceptable, with expected adverse events like cytokine release syndrome, which was managed with preplanned protocols [92].

In a phase II open-label trial called SPEARHEAD 1, D’Angelo et al. aimed to evaluate the safety, tolerability and efficacy of afamitresgene autoleucel in patients with advanced/metastatic synovial sarcoma (SS) or myxoid/round-cell liposarcoma (MLS). Afamitresgene autoleucel is a genetically engineered autologous specific peptide affinity-enhanced receptor (SPEAR) targeting MAGE-A4. Patients with MAGE-A4-expressing tumors underwent leukapheresis. Autologous T-cells were collected for processing and manufacture into afamitresgene autoleucel cells, which were infused back into the patients after lymphodepleting chemotherapy. Among 25 evaluable subjects (23 SS and 2 MLS), there were 2 CRs, 8 PRs and 11 SDs (DCR 84%). Side effects were manageable with low-grade cytokine release syndrome (CRS) and reversible hematologic toxicities (neutropenia, anemia) due to lymphodepleting chemotherapy [93]. Unlike TCRs, which can only recognize major histocompatibility complex (MHC1)-restricted peptides, CAR T-cells can target any protein expressed on the tumor cell surface. See Table 4 for ongoing trials for TCR therapy in sarcoma.

## 10. Oncolytic Viruses

Oncolytic viruses mediate antitumor activity through two distinct mechanisms of action: selective replication within neoplastic cells, resulting in a direct lytic effect on tumor cells, and the induction of systemic antitumor immunity. Tumor cell lysis releases tumor-specific antigens that trigger both the innate and adaptive immune systems [94]. Tumor antigens released by cancer cells are processed by antigen-presenting cells (APC) and presented to the CD4+ and CD8+ lymphocytes, triggering the immune response that enhances tumor destruction. OVs fall under two major categories: natural viruses and genetically modified virus strains. Natural viruses include wild-type and naturally variant strains of weak viruses [95]. With the development of genetic editing technology, these wild virus strains are optimized to weaken viral pathogenicity and improve immunogenicity. The insertion of an exogenous therapeutic gene into the OV genome makes it possible to avoid systemic immune response and enhances the lethality of the virus [96]. See Figure 4 for Oncolytic virus structure and mechanism of action.

In a study by Le Boeuf et al., four oncolytic viruses, reovirus, vaccinia virus, herpes simplex virus and two rhabdoviruses (vesicular stomatitis virus and maraba virus MG1) were screened for their ability to infect and kill sarcoma cell lines in vitro. In the in vitro setting, both rhabdoviruses were noted to be highly potent in killing sarcoma cells, with MG1 showing productive viral replication in 18 of 21 tumor samples (86%) and inducing >50% cell death at lower concentrations. Ex vivo, the efficacy of MG1 was tested on murine models infected with tumor cells that were seeded subcutaneously in mice. MG1 was then administered intra-tumorally. The results showed that MG1 effectively replicates in murine sarcoma tumors and leads to the eradication of 80% of tumors. Additionally, MG1 also induced the generation of a memory immune response that provided protection against a subsequent tumor challenge [97].

The modified herpes simplex virus, known as talimogene laherparepvec (T-VEC), was FDA-approved for the treatment of melanoma in 2015. The success of T-VEC in melanoma has led to further research into its efficacy in treating other solid malignancies. In a phase IB/II trial, Monga et al. explored a novel combination of T-VEC with external beam radiation therapy (EBRT) administered preoperatively in patients with locally advanced STS of the extremities and trunk. The combination was safe and well-tolerated; however, only 5 of the 23 evaluable patients achieved the primary endpoint of pathological complete response (pCR defined as ≥95% tumor necrosis) [98]. In another phase II clinical trial by Kelly et al., treatment with T-VEC plus pembrolizumab was associated with strong antitumor activity in advanced sarcoma across a range of sarcoma histologic subtypes, with a manageable safety profile. The study met its primary endpoint, with an ORR at 24 weeks of 35% (95% CI, 15–59%; *n* = 7) [99].

Overall, the aforementioned studies suggest that OVs could be promising immunotherapies for the treatment of sarcoma. OVs have achieved limited success as monotherapy, though they will likely require use in combination with other modalities that can overcome known resistance mechanisms, including innate antiviral responses and immunological resistance.

## 11. Cancer Vaccines

Cancer vaccines are a realm of immunotherapy where selected tumor antigens are exogenously administered along with adjuvants/immunostimulants, such as granulocyte-macrophage colony-stimulating factor (GM-CSF) or interferon-gamma a, to induce the activity of APCs, mainly dendritic cells, aiming to stimulate the adaptive immune system against cancer cells. Antigens for vaccines can be procured from: (1) killed tumor cells, (2) antigens purified from patients with tumors, and (3) antigens produced in vitro. Commonly over-expressed CTAs in sarcomas are NY-ESO-1, MAGE, PRAME (preferentially expressed antigen of Melanoma), BAGE (B melanoma antigen), and CAGE (cancer-associated antigen gene), all of which are excellent targets for vaccines. See Figure 5 for dendritic and peptide cancer vaccines.

In a randomized phase II study by Carvajal et al., an immunological adjuvant with a conjugated ganglioside vaccine targeting ganglioside monosialic (GM2), diasialoganglioside (GD2), GD3 and a control was tested in patients with metastatic sarcoma following complete metastasectomy. Patients received a total of ten injections, and imaging was performed to evaluate the response. The primary endpoint was PFS and the secondary endpoints were overall survival and serologic response. The median PFS and 1-year PFS rate were 6.4 months and 35%, respectively, with no difference between arms. The 1-year OS rate was >90%. Serologic responses (IgM and/or IgG) to GM2 and GD2 were observed in 98% and 21% of patients treated with the complete vaccine and control, respectively. No difference in PFS was observed between arms [100].

Unique chromosomal translocation events are common within certain sarcoma subtypes, such as the t(X;18)(p11;q11) translocation in synovial sarcoma or the t(12;16)(q13;p11) translocation in myxoid/round-cell liposarcoma. These translocations are attractive vaccine targets as the newly formed peptide will potentially represent a tumor-specific neoantigen. A fragment of the SYT-SSX fusion peptide that results from the characteristic synovial sarcoma translocation was studied by Kawaguchi et al. as a vaccine in 21 patients with advanced synovial sarcoma that were deemed unresectable and previously failed the first line of treatment. One out of nine patients who received the peptide fragment alone did not have disease progression within the study period, and six out of twelve patients who received the peptide with an adjuvant and interferon-α had stable disease. One patient exhibited transient shrinkage of a metastatic lesion [101].

In a phase II study, Chawla et al. studied CMB305 and atezolizumab compared with atezolizumab alone in soft-tissue sarcomas expressing NY-ESO-1 [102]. CMB305 is a vaccination regimen created to prime NY-ESO-1-specific CD8 T-cell populations and then activate the immune response with a potent toll-like receptor 4(TLR-4) agonist. Patients with locally advanced, relapsed, or metastatic synovial sarcoma or myxoid liposarcoma were randomly assigned to receive CMB305 with atezolizumab or atezolizumab alone. PFS was 2.6 months and 1.6 months in the combination and control arms, respectively (hazard ratio, 0.9; 95% CI, 0.6 to 1.3). Median OS was 18 months in both treatment arms. The combination did not result in significant increases in PFS or OS compared to monotherapy with atezolizumab alone. Some patients demonstrated evidence of an anti-NY-ESO-1 immune response and appeared to fare better by imaging than those without such an immune response; however, this combination approach merits further evaluation.

Although cancer vaccines for sarcoma appear to be safe and result in an immunological response in most of patients, the clinical outcomes of patients are limited, which suggests that many modifications need to be made to attain better therapeutic outcomes. Further research in this field is warranted. See Table 5 for ongoing trial for cancer vaccine therapy in sarcoma.

## 12. Cancer Targeted Antibodies

Gangliosides are plasma membrane-bound glycosphingolipids that interact with membrane proteins to regulate the cell signaling pathway [103,104,105]. The monosaccharide component protruding outside of the cell membrane has antigenic properties and participates in intercellular communication and adhesion [105,106,107]. Multiple subtypes of gangliosides, such as GM3, GM2, and GM1, are found on normal cells and regulate the function of membrane-bound signaling proteins [107,108]. However, disialoganglioside (GD2) is expressed mostly on tumor cells, with limited expression on normal central and peripheral nerve fibers, mesenchymal stem cells, melanocytes, and lymphocytes [109,110]. This specific tumor antigenic quality of GD2 becomes not only an interesting target in cancer immunotherapy but also a biomarker to predict prognosis and a cancer imaging modality via radioimmunodetection [111,112].

GD2 expression is notable in Ewing sarcoma, usually confirmed by immunostaining [113,114]. In osteosarcoma, a higher intensity of IHC staining was observed in recurrent or relapsed disease tissue sections compared to the initial tissue resection [115]. A combination therapy of anti-GD2 mAb (14G2a) and cisplatin has a synergistic effect on the apoptosis of the osteosarcoma cells in vitro [116]. In the study, 70–85% of cell apoptosis was observed in osteosarcoma cells treated with a cisplatin and 14G2a combination. In soft-tissue sarcoma, the expression of GD2 varies from 25% to 93% among different subtypes [117,118].

Another interesting target is CD47, a transmembrane-bound protein highly expressed in some tumor cells, including angiosarcomas. By producing CD47, tumor cells resist phagocytosis by macrophages; as such, inhibiting CD47 could result in increased tumor cell death [119]. In one of the vitro studies, anti-CD47 therapy increased the production of pro-inflammatory cytokines in the TME of soft-tissue sarcomas [120]. In an in vivo study using a murine model, the combination of the anti-GD2 antibody dinutuximab and an anti-CD47 antibody (B6H12) was shown to have synergistic activity [121]. In this study, mice with osteosarcoma with pulmonary metastases were treated with a control antibody, anti-GD2, anti-CD47, or a combination of both anti-GD2 and anti-CD47. It was found that the anti-GD2 antibody alone did not alter the burden of pulmonary metastases, the anti-CD47 antibody alone reduced the burden of metastases, and the combination treatment eradicated nearly all pulmonary metastatic disease [121]. This is in keeping with a previous trial where dinutuximab (anti-GD2) was used as a single agent in relapsed osteosarcoma in children and young adults, in which the disease control rate did not improve [122]. The reasoning for the combination being more potent is a synergism in which anti-GD2 primes tumor cells for phagocytosis via the upregulation of surface proteins, while anti-CD47 prevents the tumor’s “don’t eat me” signals [121]. An ongoing phase I clinical trial (NCT04751383) is testing the combination therapy of magrolimab (anti-CD47) and dinutuximab (anti-GD2) in patients with relapsed or refractory neuroblastoma or relapsed osteosarcoma.

## 13. Conclusions

The impact of modern immunotherapeutic modalities across various cancer types presents an exciting opportunity for further studies in the treatment of sarcomas. An accumulating understanding of the immune microenvironment and antigenic signatures of various sarcoma subtypes has generated promising new targets for immunotherapy. Despite unrivaled progress in the field of immune oncology over the last decade, early experiences of immunotherapy with sarcomas have been disappointing due to antigenic heterogeneity and the rarity of the disease. The antigenic heterogeneity and rarity of this disease also make it challenging to enroll a statistically significant number of patients in clinical trials to achieve any substantial results. For future clinical trials, the investigators should consider categorizing the sarcoma subtypes into immunologically “hot” or “cold” tumors based on overall tumor immunogenicity and/or the presence or absence of suppressor cells in the TME. Although it is difficult to adequately capture the complexity of sarcomas, it appears that combination therapies involving ICBs are likely the path forward. When it comes to sarcomas, there is no “one size fits all” strategy, and each subtype will require a stringent characterization of its immune components and antigenic signatures to select an optimal treatment modality. Further studies are encouraged to develop effective immunotherapy-based regimens for the treatment of sarcomas and to produce better responses and clinical outcomes with manageable toxicity profiles.

## Figures and Tables

**Figure 1 ijms-25-01266-f001:**
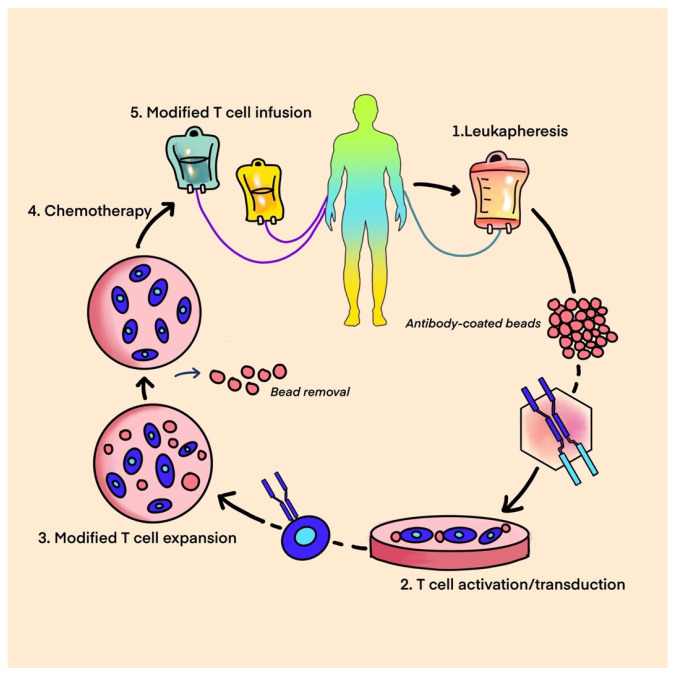
CAR T-cell-Leukapheresis, T-cell modification, expansion, and CAR T-cell infusion.

**Figure 2 ijms-25-01266-f002:**
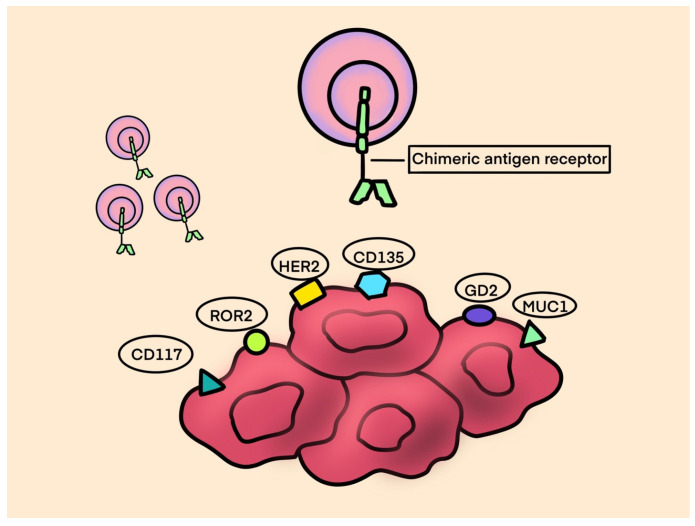
CAR T-cell structure.

**Figure 3 ijms-25-01266-f003:**
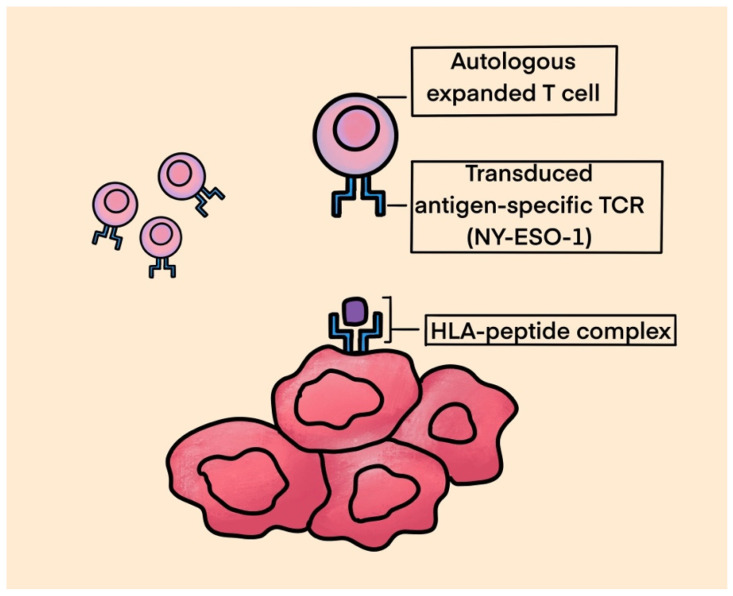
TCR.

**Figure 4 ijms-25-01266-f004:**
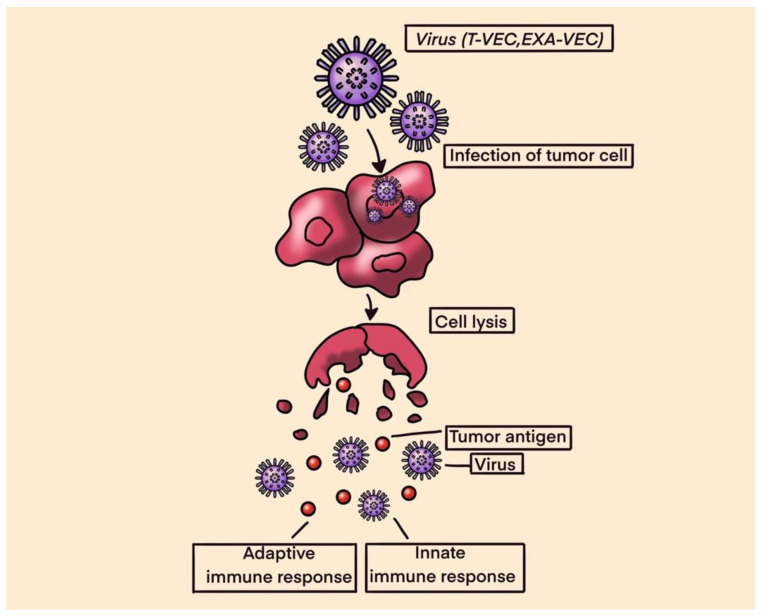
Oncolytic Viruses.

**Figure 5 ijms-25-01266-f005:**
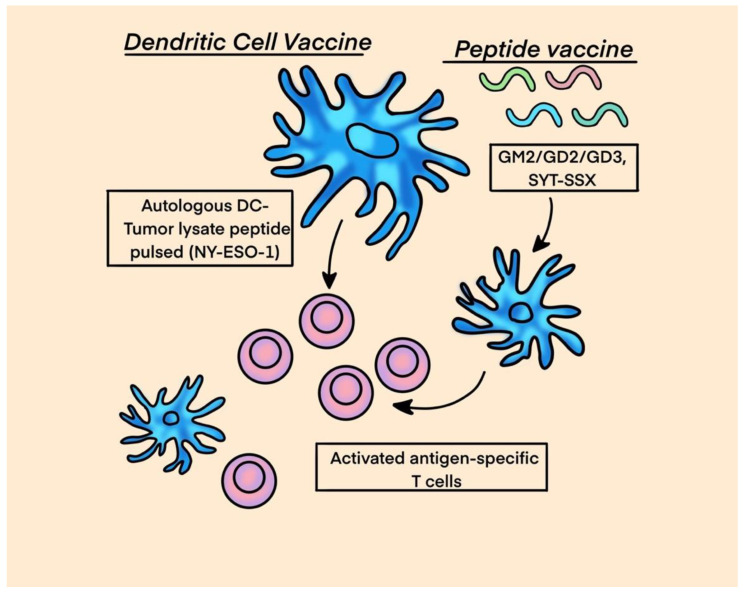
Cancer Vaccines.

**Table 1 ijms-25-01266-t001:** Results of Selected Trials of Immunotherapy in Sarcoma.

Clinical Trial/Design	Phase	Agent/ Intervention	Indication/Prior Lines of Treatment	Evaluated Patients (*n*) and Tumor Subtypes	ORR (%)	PFS (Weeks (w) or Months (m))	OS (w or m)	Outcomes in Subtypes/Notes
**ICI Monotherapy or Combination**								
**Maki et al.,** **2013** **[31]**	Phase II	ipilimumab	Locally recurrent or metastatic SS, at least 1 prior line treatment	6 SS	0%	1.85 m	8.75 m	
**Tawbi et al.** **(SARC028)** **2017** **[30]**	Phase II	pembrolizumab	Advanced/metastatic STS or bone sarcoma, at least 1 prior line treatment	40 BS cohort (22 OS, 13 ES, 5 CS)	-	8 w	52 w	1 PR in CS and 1 PR in OST
				40 STS cohort (10 LMS, 10 LPS, 10 SS, 10 UPS)	-	18 w	49 w	1 CR and 3 PRs in UPS, <2 PRs in LPS, 1 PR in SS
**Ben-Ami et al.,** **2017** **[32]**	Phase II	nivolumab	Advanced or metastatic uterine LMS, at least 1 prior line of treatment	12 LMS	0%	1.8 m	-	
**D’Angelo et al.** **Alliance A091401** **2018** **[38]**	Phase II	nivolumab/ipilimumab vs. nivolumab	Advanced or metastatic BS and STS, at least 1 prior line of treatment	Nivolumab/ipilimumab: 42 (3 AS, 4 BS, 14 LMS, 2 LPS, 6 SCS, 2 SS, 6 UPS/MFH, 1 unspecified sarcoma, 4 others)	16%	4.1 m	10.7 m	Response in uterine LMS, non-uterine LMS, MFS, UPS/MFH, AS
				Nivolumab: 43 (5 BS, 15 LMS, 3 LPS, 2 unspecified sarcoma, 5 SCS, 2 SS, 5 UPS, 6 others	5%	1.7 m	10.7 m	1 PR in ASPS and 1 PR in non-uterine LMS
**Uboha et al.,** **2019** **[39]**	Phase II	spartalizumab + LAG525 (anti-LAG3)	Advanced solid tumors and hematologic malignancies	10	CBR 40%	-	-	Sarcoma cohort did not meet the expansion criterion
**Zhou et al.,** **2020** **[42]**	Retrospective	nivolumab + ipilimumab	Advanced or metastatic STS, 87% received at least 1 prior line	38 (9 LMS, 8 Sarcoma NOS, 6 LPS, 5 MFS, 3 MPNST, 2 SFT, 1 Breast AS, 1 FDFP, 1 RMS, 1 SS)	15%	2.7 m	12 m	CR in 1 MFS, 1 PR in each MPNST, SFT, MFS, DDLS, and sarcoma NOS
**Naqash et al.,** **2021** **[40]**	Phase II	pembrolizumab	Advanced or metastatic ASPS	43	37.2%	-	-	
**Blay et al.** **French AcSé** **2021** **[34]**	Phase II	pembrolizumab	Advanced rare sarcoma	98 (34 chordoma, 14 ASPS, 11 SMRT, 8 DSCRT, 31 other histotypes)	15.3	2.75 m	19.7 m	Highest ORR in chordoma, ASPS, SMRT, DSCRT
**Delyon et al.,** **2022** **[43]**	Phase II	pembrolizumab	Classic/endemic Kaposi sarcoma with extensive cutaneous extension, 71% had at least 1 prior line	17 (8 classic KS and 9 endemic KS)	71%	-	-	
**Zer et al.,** **2022** **[44]**	Phase II	ipilimumab and nivolumab	Classic Kaposi sarcoma, at least 1 prior line of treatment	11	45%	not reached	-	
**Somaiah et al.,** **2022** **[45]**	Phase II	durvalumab + tremelimumab.	Advanced or metastatic sarcoma (BS and STS), 91% had at least 1 prior line	57 (3 DDLPS, 2WDLPS, 1 PLS, 5 AS, 5 LMS, 5 UPS, 5 SS, 1 CDOS, 4 COS, 10 ASPS, 5 chordomas, 11 other sarcomas)	12%	2.8 m	21.6 m	ASPS ORR 40%
**ICI Combination with TKI**								
**Schoffski et al.,** **2016** **[46]**	Ia/Ib	pembrolizumab + olaratumab (monoclonal antibody against platelet derived growth factor receptor alpha)	Advanced or metastatic STS, 92% had at least 1 prior line	28	21.4%	2.7 m	14.8 m	
**Paoluzzie et al.,** **2016** **[47]**	Retrospective study	durvalumab + pazopanib	Metastatic STS and BS, median 2 prior lines of treatment	28 (24 STS, 4 BS with 24 evaluable patients)	10%	-	-	3 PRs (1 DDCS with nivolimab alone), 1 EpS, 1 MOS)
**Wilky et al.,** **2019** **[48]**	Phase II	pembrolizumab + axitinib	Advanced or metastatic STS, 81% with at least 1 prior line of treatment	33 (12 ASPS, 6 LMS (4 uterine), 5 High-grade PS, 2 DDLPS, 8 other histotypes)	25%	4.7 m	18.7 m	ASPS ORR 50%
**Xie et al.** **APFAO trial** **2020** **[49]**	Phase II	camrelizumab + apatinib	Advanced or metastatic OS, at least 1 prior line of treatment	43 (OS including osteoblastic, chondroblastic, fibroblastic and small cell)	20.1%	6.2 m	11.3 m	
**Palmerini et al.** **IMMUNOSARC** **2020** **[50]**	Phase II	nivolumab + sunitinib	Advanced BS cohort, at least 1 prior line of treatment	40 (17 OS, 14 CS, 8 ES, 1 bone UPS, 4 DDCS)	5%	3.7 m	14.2 m	1 CR in DDCS and 1 PR in OS
**Martin-Broto et al.** **IMMUNOSARC** **2020** **[51]**	Phase I/II	nivolumab + sunitinib	Metastatic STS, at least 1 prior line of treatment	52 (9 SS, 8 UPS, 7 clear cell sarcoma, 7 SFT, 7 EpS, 5 AS, 4 ESMCS, 4 ASPS, 1 EHET)	21%	5.6 m	-	1 CR in AS, 2 PRs in ASPS, 1 PR in ESMCS and 1 PR in SS
**Kim et al.,** **2021** **[52]**	Phase II	durvalumab + pazopanib	Advanced or metastatic STS, at least 1 prior line of treatment	46	28.3%	7.7 m	-	Objective responses in ASPS, AS, UPS, DSRCT
**Cousin et al.** **REGOMUNE** **2022** **[53]**	Phase II	avelumab + regorafenib	Advanced or metastatic STS, at least 1 prior line of treatment	43 (22 LMS, 9 SS, 4 LPS, 4 UPS, 10 other subtypes)	9.3%	1.8 m	15.1 m	
**Allred et al.** **Alliance A091902 trial** **2023** **[54]**	Phase II	nivolumab with carbozantinib	Advanced AS, previously treated	18 (AS including 12 cutaneous, 1 liver, 2 breast, 6 others)	72%	9.6 m	20.5 m	
**Eulo et al.,** **2023** **[55]**	Phase II	nivolumab/ipilimumab + cabozantinib N/I + C	Metastatic STS that lacks translocation, at least 1 prior line of treatment	69 (N/I + C arm)	11%	5.4 m	-	
				36 (C only arm)	6%	3.8 m	-	
**ICI Combination with Chemotherapy**								
**Toulmonde et al.,** **2018** **[56]**	Phase II	pembrolizumab + metronomic cyclophosphamide	Advanced or metastatic STS, 97% with at least 1 prior line of treatment	50 (15 LMS, 16 UPS, 16 other sarcomas, 10 GIST)	2%	1.4 m	-	
**Italiano et al. Amended PEMBROSARC** **2022** **[23]**	Phase II	pembrolizumab with metronomic cyclophosphamide	TLS-positive advanced STS, 63% had at least 1 prior line of treatment	35 (12 WDLPS/DDLPS, 4 LMS, 6 UPS, 3 EpS, 10 other histotypes)	30%	6 m PFS 40%	-	PRs: 5 in DDLPS, 3 EpS, 1 LMS SDs: 6 DDLPS, 1 FMS, 1 MFS, 1 uterine LMS, 1 UPS
**Nathenson et al.,** **2020** **[57]**	Phase II	pembrolizumab + eribulin	Metastatic STS, at least 1 prior line of treatment	19 LMS (11 uterine LMS)	5.3%	11.1 w	-	
**Smrke et al.,** **2021** **[58]**	Phase I	pembrolizumab + gemcitabine	Advanced or metastatic LMS, UPS	13 (2 UPS, 11 LMS)	-	5.1 m	-	LMS—DCR 73% (8 SDs, 3 PDs) UPS—DCR 100% (2 PRs)
**Wagner et al.,** **2022** **[59]**	Phase I/II	avelumab + trabectedin	Advanced or metastatic LPS and LMS, 86% had at least 1 prior line of treatment	35, only 23 evaluable (24 with LMS, 11 with LPS	13%	8.3 m	27 m	LMS 4 PRs, 9 SDs LPS 7 SDs
**Toulmonde et al.,** **2022** **[60]**	Phase Ib	durvalumab + trabectedin	Advanced or metastatic STS cohort, at least 1 prior line of treatment	16 (6 LMS, 2 DDLPS, 8 others)	7%	12 m PFS 14.3%	-	
**Adnan et al.** **Gallant trial** **2022** **[61]**	Phase II	nivolumab + metronomic gemcitabine, doxorubicin, and docetaxel	Advanced or metastatic STS, at least 1 prior line of treatment	39 (15 LMS, 4 PS, 4 SS, 3 LPS, 3 OS, 10 others)	20.5%	4.6 m	6.2 m	mPFS 2 m historically in previously treated patients
**Andreou et al.** **NITRA-SARC** **2023** **[62]**	Phase II	nivolumab + trabectedin	Advanced or metastatic STS, at least 1 prior line of treatment	Group A—43 (28 LMS and 15 LPS)	-	5.5 m	18.7 m	
				Group B—49 (12 UPS, 11 SCS, 6 FMS, 5 SS, 4 EpS)	-	2.3 m	5.6 m	
**Beveridge et al.** **ImmunoSarc2** **Cohort 7b** **2023** **[63]**	Phase Ib	doxorubicin and dacarbazine plus nivolumab and nivolumab maintenance 1 year	Advanced or metastatic LMS, anthracycline naïve patients	16 LMS	56%	8.67 m	-	
**ICI as Front-line**								
**Pollack et al.,** **2020** **[64]**	Phase I/II	pembrolizumab + doxorubicin	Anthracycline naïve sarcoma Excluding ES, ARMS, ERMS, 76% with no prior line of treatment	37 (11 LMS, 4 DDLPS, 3 CCCS, 3 UPS, 2 SFT, 2 ESS, 2 EHET, 8 other histotypes	19%	8.1 m	27.6 m	
**Livingston et al.,** **2021** **[65]**	Phase II	pembrolizumab + doxorubicin	Anthracycline Naïve advanced STS, 86.7% had no prior treatment	28 (7 LPS, 10 LMS, 1 SS, 4 UPS, 2 AS, 6 other histotypes)	36.7%	5.7 m	17 m	The most clinical benefit in UPS, EpAS, LMS, LPS
**Maleddu et al.,** **2023** **[66]**	Phase II	doxorubicin + anti-CTLA-4 zalifrelimab and anti-PD1 balstilimab	Advanced or metastatic STS, no prior doxorubicin or ICI	28	36%	25.6 m	-	Responses seen in IS, AS, MPNST, LPS, LMS, ESS, UPS, and SEpF.
**Gordon et al.** **SAINT Trial** **2023** **[67]**	Phase I/II	ipilimumab + nivolumab and trabectedin	Advanced or metastatic STS, treatment naive	101 (14 LPS, 26 LMS, 9 UPS, 7 RMS, 5 SS, 4 clear CS, 4 PS, 4 MFS, 3 PNST, 3 MLS, 2 carcinosarcoma, 2 DSRCT, 2 sarcoma NOS)	25.3%	6.7 m	24.6 m	
**Chen et al.,** **2021** **[36]**	Retrospective	nivolumab + ipilimumab vs. nivolumab	Metastatic STS (100% PD-L1-positive tumors: PD-L1 expression > 1%), treatment naive	74—Nivolumab and ipilimumab arm 43 non-uterine LMS, 20 LPS, 11 SS)	-	4.1 m	12.2 m	
				76—Nivolumab arm (40 non-uterine LMS, 22 LPS, 14 SS)	-	2.2 m	9.2 m	

Alveolar rhabdomyosarcoma—ARMS; Alveolar soft part sarcoma—ASPS; Angiosarcoma—AS; Desmoplastic small round-cell tumor—DSRCT; Dedifferentiated chondrosarcoma—DDCS; Epithelioid sarcoma—EpS; Extraskeletal myxoid chondrosarcoma—ESMCS; Breast angiosarcoma—breast AS; Chondrosarcoma—CS; Chondroblastic osteosarcoma—CDOS; Clear cell chondrosarcoma—CCCS; Clear cell sarcoma—Clear CS; Conventional osteosarcoma—COS; Dedifferentiated liposarcoma—DDLPS; Embryonal rhabdomyosarcoma—ERMS; Epithelioid Angiosarcoma—EpAS; Endothelial stromal sarcoma—ESS; Epithelioid hemangioendothelioma—EHET; Epithelioid sarcoma—EpS; Fibromyxoid sarcoma—FMS; Fibrosarcomatous dermatofibrosarcoma protuberans—FDFP; Gastrointestinal stromal tumor—GIST; High-grade pleomorphic sarcoma—High-grade PS; Intimal sarcoma—IS; Pleomorphic liposarcoma—PLS; Pleomorphic sarcoma—PS; Malignant fibrous histiocytoma—MFH; Malignant peripheral nerve sheath tumor—MPNST; Maxillary osteosarcoma—MOS; Myxofibrosarcoma—MFS; Leiomyosarcoma—LMS; Liposarcoma—LPS; Osteosarcoma—OS; Rhabdomyosarcoma—RMS; Sclerosing epithelioid fibrosarcoma—SEpF; Spindle cell sarcoma—SCS; Smarca4-deficient malignant rhabdoid tumor—SMRT; Solitary fibrous tumor—SFT; Synovial sarcoma—SS; Undifferentiated pleomorphic sarcoma—UPS; Well-differentiated liposarcoma—WDLPS. ORR—Objective response rate; OS—Overall survival; PFS—Progression-free survival.

**Table 2 ijms-25-01266-t002:** Currently Ongoing Selected Clinical Trials for Immunotherapy in Sarcomas.

Phase	NCT Number/Trial Name	Status	Conditions	Interventions
** ICI **				
Phase I/II	NCT03138161 SAINT	Recruiting	Unresectable or metastatic STS as first-line treatment	trabectedin + ipilimumab + nivolumab
Phase II	NCT04095208	Recruiting	Advanced or Metastatic STS (TLS+)	nivolumab + relatimab vs. nivolumab
Phase II	NCT04802876 ACROPOLI (SOLTI-1904)	Recruiting	Across multiple cancer types with PD1-high mRNA Expressing Tumors—Include Sarcoma Cohort	spartalizumab + tislelizumab
** ICI with TKIs **				
Phase II	NCT04784247	Recruiting	Advanced STS	lenvatinib + pembrolizumab
Phase II	NCT05182164	Recruiting	Advance sarcomas: ES, OS, UPS	pembrolizumab + carbozantinib
Phase II	NCT04551430	Active, not recruiting	Metastatic STS	cabozantinib + nivolumab + ipilimumab
** ICI with Chemotherapy **				
Phase II	NCT03899805	Active, not recruiting	STS (LPS, LMS, UPS)	eribulin + pembrolizumab
Phase II	NCT04535713 GALLANT	Recruiting	Advanced sarcoma	metronomic gemcitabine + doxorubicin + docetaxel + nivolumab
Phase I/II	NCT05876715 LINNOVATE	Recruiting	Advanced STS	lurbinectedin + nivolumab + ipilimumab
Phase I/II	NCT04577014	Recruiting	Advanced STS	retifanlimab + gemcitabine + docetaxel
Phase II	NCT04028063	Recruiting	Advanced STS	doxorubicin + zalifrelimab − AGEN1884 + balstilimab − AGEN2034
** ICI with Radiation Therapy **				
Phase I	NCT05488366	Recruiting	Metastatic STS	pembrolizumab + Radiation Therapy
Phase II	NCT03307616	Active, no recruiting	Recurrent or resectable DDLPS and UPS before surgery	Nivolumab ± ipilimumab + Radiation Therapy
Phase I/II	NCT03116529	Active, not recruiting	High-risk STS	durvalumab + tremelimumab + Radiation + Surgery

Retrieved from www.clinicaltrials.gov, accessed on 6 October 2023.

**Table 3 ijms-25-01266-t003:** Ongoing clinical trials for CAR T-cell therapy in Sarcoma.

Trial Number	Phase	Intervention	Disease
**NCT01953900**	Phase I	Anti-GD2 T-cells in combination with a varicella zoster vaccine and lymphodepleting chemotherapy	GD2-positive sarcoma and neuroblastoma in relapsed or refractory setting
**NCT04995003**	Phase I	Anti-HER2 CAR T-cells in combination with an immune checkpoint inhibitor drug (pembrolizumab or nivolumab)	HER 2-positive Sarcoma in patients disease progression or recurrence after at least one prior systemic therapy
**NCT02107963**	Phase I	Administering escalating doses of autologous anti-GD2-CAR T-cells	Osteosarcoma, GD2+ solid tumors that recurred or progressed on treatment
**NCT00902044**	Phase I	Anti-HER2 CAR T-cells with fludarabine and cyclophosphamide	Refractory HER2-positive sarcoma or metastatic HER2-positive sarcoma with disease progression after receiving at least one prior systemic therapy
**NCT03721068**	Phase I	Anti-GD2 CAR T-cells, fludarabine and cyclophosphamide	Relapsed refractory osteosarcoma and neuroblastoma

**Table 4 ijms-25-01266-t004:** Ongoing Trials for TCR Therapy in Sarcoma.

Clinical Trial	Phase	Intervention	Disease
NCT03462316	Phase I	Anti-NY-ESO-1 (TCR Affinity-Enhancing Specific T-cell Therapy)	Advanced bone and soft-tissue sarcoma that failed first-line treatment
NCT05296564	Phase I	Anti-HBI 0201-ESO TCRT (anti-NY-ESO-1 TCR-Gene Engineered Lymphocytes)	NY-ESO-1—Expressing Metastatic cancers (synovial sarcoma, STS, etc.) that failed first-line or second-line treatment, recurrence of disease, progression of disease
NCT03132922	Phase I	Genetically Engineered Anti-MAGE-A4	MAGE-A4-Positive Tumors (synovial sarcoma, myxoid round-cell liposarcoma) failed first line of therapy

**Table 5 ijms-25-01266-t005:** Ongoing Trial for Cancer Vaccine Therapy in Sarcoma.

NCT	Phase	Intervention	Disease
NCT01241162	Phase I	Mature DC pulsed with peptides derived from NY-ESO-1, MAGE-A1, and MAGE-A3 for vaccine production.	Relapsed refractory Ewing sarcoma, osteogenic sarcoma, rhabdomyosarcoma or synovial sarcoma

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
