# Peer review of "Role of Immunotherapy in Sarcomas"

_ijms, 2024, doi:10.3390/ijms25021266_

Round 1
Reviewer 1 Report
Comments and Suggestions for Authors
This is an excellent review that is both timely and well-organized. This reviewer only has a comment that could be added if the authors choose to do so to the discussion. My comment is that immunotherapy protocols need not and should not be organized as those for other types of therapy such as chemotherapy and radiation therapy. This is especially true for rare tumors and and those, like sarcomas, that consist of so many subtypes. Hence, instead of basing clinical trials on histologic subtype, it is recommended that clinical investigators base patient selection on the immune profile of each patient's tumor. Hence, criteria that should be considered are: "cold" vs "hot" tumors, expression of HLA and the overall immunogenicity of tumors indicated by the influx of immune cells in the tumor microenvironment (TME), and the presence or absence of suppressor cells in the TME. In this way, patients with various sarcomas could be organgized base on the immunological profile and can therefore be grouped to get increased numbers per arm.
If you look at the list of trials completed or underway, all are based upon histological type and do not consider these critical issues of how the tumors defeated the immune system of the patient.
Author Response
Respected reviewer,
I appreciate your time and effort in reviewing this manuscript on Role of immunotherapy in Sarcoma. I have incorporated your excellent suggestion on immunologically “hot” and “cold” tumors and the need for designing clinical trials based on immunogenicity of sarcomas instead of specific sarcoma subtypes under the conclusion section of this review article.
Thank you

Reviewer 2 Report
Comments and Suggestions for Authors
This is a well-written review article on the status of the development of immunotherapy for sarcoma. These are minor points, but I have a few opinions.
#1. I don't think Tables 1 and 2 are necessary. Also, if those Tables are essential, the reference cited should be the WHO book.
#2. The term "synovial cell sarcoma" is used on p.18, but "synovial sarcoma" is the correct term. This should be corrected.
#3. A few essential recent references regarding NY-ESO-1-specific TCR T cell therapy seem missing. Please consider adding the following reference.
- Safety and Efficacy of NY-ESO-1 Antigen-Specific T-Cell Receptor Gene-Transduced T Lymphocytes in Patients with Synovial Sarcoma: A Phase I/II Clinical Trial. Kawai A, et al. Clin Cancer Res. 2023 Dec 15;29(24):5069-5078. doi: 10.1158/1078-0432.CCR-23-1456.
- Phase 1 clinical trial to assess safety and efficacy of NY-ESO-1-specific TCR T cells in HLA-A∗02:01 patients with advanced soft tissue sarcoma. Pan Q, et al. Cell Rep Med. 2023 Aug 15;4(8):101133. doi: 10.1016/j.xcrm.2023.101133.
#4. References 125, 126, and 127 in the list of references appear to be erroneous. They need to be corrected.
Comments on the Quality of English LanguageMinor editing of English language required.
Author Response
Respected reviewer,
Thank you for your time and effort in thoroughly evaluating this paper on Role of immunotherapy in sarcomas.
- Table 1 and 2 have been removed along with their references
- The term synovial cell sarcoma has been replaced by synovial sarcoma.
- NY-ESO papers that you suggested have been added under the section TCR and references have been updated (92, 93) highlighted in paper
- References 125, 126 and 127 have been removed.

Reviewer 3 Report
Comments and Suggestions for Authors
In this review, the Authors aimed to discuss the biological basis, current clinical trials, and future challenges of immunotherapy in advanced sarcomas.
The topic is interesting. Please acknowledge the narrative nature of this review.
Table 1 and 2 are unuseful. Please remove.
Table 3 is confusing. Please check.
Would shortened and resume the results of different trial.
Author Response
Respected reviewer,
Thank you for your time and effort in thoroughly evaluating this paper on Role of immunotherapy in sarcomas.
- Table 1 and 2 have been removed as per your recommendation.
Reviewer 4 Report
Comments and Suggestions for Authors
This review comprehensively covers currently available information and updates related to the possibilities and challenges in the application of immunotherapy for treating patients with sarcomas in a clear and concise manner.
Comments on the Quality of English LanguageMinor editing required.
Author Response
Respected reviewer,
Thank you for your time and effort in thoroughly evaluating this paper on Role of immunotherapy in sarcomas. I appreciate your feedback.